# Ionised and total hypocalcaemia in pregnancy: An analysis of prevalence and risk factors in a resource-limited setting, Cameroon

**Atem Bethel Ajong** [1,2☯]*, **Bruno Kenfack** [3‡], **Innocent Mbulli Ali** [2‡], **Martin Ndinakie Yakum** [4‡], **Ukaogo Prince Onydinma** [5‡], **Fulbert Nkwele Mangala** [6,7‡], **Loai Aljerf** [8,9‡], **Phelix Bruno Telefo** [2☯]*

1 Kekem District Hospital, Kekem, West Region, Cameroon, 2 Department of Biochemistry, University of Dschang, Dschang, West Region, Cameroon, 3 Department of Obstetrics / Gynaecology and Maternal Health, Faculty of Medicine and Pharmaceutical Sciences, University of Dschang, Dschang, West Region, Cameroon, 4 Department of Epidemiology and Biostatistics, School of Medical and Health Sciences, Kesmonds International University, Bamenda, Cameroon, 5 Department of Pure and Industrial Chemistry, Abia State University, Uturu, Nigeria, 6 Faculty of Medicine and Pharmaceutical Sciences, University of Douala, Douala, Cameroon, 7 Maternity Unit, Nkongsamba Regional Hospital, Nkongsamba, Littoral Region, Cameroon, 8 Department of Basic Sciences, Faculty of Dentistry, Damascus University, Damascus, The Syrian Arab Republic, 9 Key Laboratory of Organic Industries, Department of Chemistry, Faculty of Sciences, Damascus University, Damascus, The Syrian Arab Republic

☯ These authors contributed equally to this work.
‡ BK, IMA, MNY, UPO, FNM and LA also contributed equally to this work.
* christrah@yahoo.fr (ABA); bphelix@yahoo.co.uk (PBT)

## Abstract

### Introduction

Hypocalcaemia remains a prevalent laboratory finding in pregnancy, capable of inducing adverse maternofoetal outcomes. This study compares the prevalence of hypocalcaemia in apparently healthy pregnant women from the ionised, and total calcaemia viewpoints and further identifies factors associated with total crude and ionised hypocalcaemia in pregnancy.

### Methods

A hospital-based cross-sectional study was conducted between November 2020 and September 2021, targeting apparently healthy pregnant women received in late pregnancy in four maternities in the Nkongsamba Health District, Cameroon. Blood samples were collected and analysed for serum ionised calcium concentrations and pH (by ion-selective electrode potentiometry), and for total calcium and albumin concentration (by atomic absorption spectrophotometry). Sociodemographic, obstetric and nutritional data were collected using an interviewer-administered questionnaire.

### Results

The average age of the 1074 participants included in the study was 28.20±6.08 years. The prevalence of total crude and total albumin-corrected hypocalcaemia in this study was 61.64

This is a Registered Report and may have an associated publication; please check the article page on the journal site for any related articles.

**Data Availability Statement:** All relevant data are within the paper and its Supporting information file.

**Funding:** The author(s) received no specific funding for this work.

**Competing interests:** The authors have declared that no competing interests exist.

[58.69–64.50]% and 56.70 [53.72–59.64]%, respectively (*p*-value = 0.000). The prevalence of ionised hypocalcaemia was very low (2.89 [2.04–4.07]%) compared with the prevalence of total hypocalcaemia (*p*-value = 0.000). Monthly income below 100.000FCFA (179 USD) (AOR = 0.73, *p*-value = 0.024), taking more than 2 meals daily (AOR = 0.68, *p*-value = 0.006) and taking desserts (AOR = 0.73, *p*-value = 0.046) reduced the odds of total crude hypocalcaemia, while having banana/plantain and tubers as the content of their most consumed meal significantly increased the odds of total crude hypocalcaemia (AOR = 1.37, *p*-value = 0.012). Single women (AOR = 2.54, *p*-value = 0.021), with a higher education (AOR = 3.27, *p*-value = 0.017), who initiated antenatal care before 4 months (AOR = 2.47, *p*-value = 0.029), had their odds of ionised hypocalcaemia significantly increased. On the other hand, women below 30 years (AOR = 0.44, *p*-value = 0.039), with occupations other than housewife (AOR = 0.34, *p*-value = 0.027), and women who took desserts between meals (AOR = 0.45, *p*-value = 0.034) had their odds of ionised hypocalcaemia significantly reduced. Taking calcium supplements simultaneously with other supplements also significantly reduced the odds of total hypocalcaemia in pregnancy (OR = 0.69, *p*-value = 0.027).

## Conclusion

Ionised hypocalcaemia in pregnancy is a rare finding. Only 2.89% of all apparently healthy pregnant women have ionised hypocalcaemia in late pregnancy, while 56.70% have total hypocalcaemia. Factors like the daily number of meals, taking of desserts, the content of the most consumed meal and monthly revenue significantly affect the prevalence of total hypocalcaemia in pregnancy. On the other hand, factors like age above 30 years, having a higher education, being single, having initiated antenatal care before 4 months of pregnancy, being a housewife and not taking desserts between meals have a significantly positive association with ionised hypocalcaemia.

## Introduction

Hypocalcaemia in pregnancy remains a prevalent finding combatted by the current World Health Organisation (WHO) recommendations on calcium supplementation. According to evidence from cross-sectional surveys, the prevalence of total hypocalcaemia in the third trimester of pregnancy is very high, with variations depending on the setting. In India and Algeria, the third-trimester prevalence of total hypocalcaemia was 66% [1] and 70% [2], respectively. In Cameroon, a recent study at the Nkongsamba Regional Hospital (NRH) reported a total hypocalcaemia prevalence of 59% [3]. In Nigeria, the prevalence of total hypocalcaemia in pregnancy (including women from all trimesters, with a higher cut-off of 88mg/L) was still as high as 29% [4]. According to a small-scale Nigerian study, this prevalence of total hypocalcaemia increases with gestational age, passing from 25.6% in the first two trimesters to 40% in the third trimester [5]. All the above studies measured total calcaemia and not ionised fractions of calcium in blood.

It is argued that total hypocalcaemia in pregnancy, even though a common finding, can be ignored because ionised calcium concentrations remain sensibly within normal range throughout pregnancy [6]. This has discouraged studies aimed at measuring the burden of ionised hypocalcaemia in pregnancy. Notwithstanding, growing evidence is associating

hypocalcaemia in pregnancy and low calcium intake to hypertensive diseases in pregnancy, a leading cause of maternal mortality worldwide. Low calcium intake increases the likelihood of foetal growth restriction, foetal low bone mass, low birth-weight and shorter birth length [6].

Insufficient calcium intake before and during pregnancy has been reported in systematic reviews to characterise developing countries like Cameroon [7, 8]. The high prevalence of hypocalcaemia in Cameroon could be associated with nutritional factors and failure to supplement. Despite recommendations from the WHO for systematic supplementation with calcium during pregnancy in regions with insufficient calcium intake (principally to prevent hypertensive diseases in pregnancy and their complications) [9], the recent study carried out in Cameroon found out that 43% of women went through pregnancy without taking calcium supplements [3].

Despite the risk associated with hypocalcaemia in pregnancy, very few studies have tried to identify the risk factors associated with this imbalance in pregnancy. In one Malaysian study, factors like low milk intake in pregnancy, underweight and obesity before pregnancy were found to be associated with an increased likelihood of hypocalcaemia in pregnancy [10]. Another study in Nigeria found that primigravidity, anaemia in pregnancy, anorexia and a low level of education significantly increased the occurrence of hypocalcaemia in pregnancy [4].

Apart from calcium supplementation, a great deal of calcium requirements are met in our diets [8, 11]. Due to food unavailability, inaccessibility, and wastage, Cameroon is classified as a country with food insecurity. According to the World Food program, 16% of Cameroonian households suffer from food insecurity [12]. Results from a study in Yaoundé, Cameroon, show that only 38% of women knew they should have at least 3 meals a day while only 22% actually practiced this [13]. Fruit and vegetable consumption have been reported to have beneficial effects on calcium stores and bone mineral density [14, 15]. As far as fruit consumption and availability is concerned, pineapples, watermelons, and oranges are classified to be most accessible and commonly consumed [12].

To the best of our knowledge no study in Cameroon has been designed to measure the burden of ionised hypocalcaemia in pregnancy, and very little interest has been put on sociodemographic, nutritional and obstetric factors affecting calcaemic states in pregnancy. Moreover, sociodemographic, nutritional and obstetric factors associated with hypocalcaemia in pregnancy are still understudied in Sub-Saharan Africa. Globally, no study has measured the prevalence of hypocalcaemia in pregnancy from the ionised calcium viewpoint. This work compares the prevalence of ionised hypocalcaemia in apparently healthy pregnant women to the total hypocalcaemia prevalence. It also presents the sociodemographic, obstetrical and nutritional factors associated with ionised and total hypocalcaemia in pregnancy among a sample of pregnant women in the Nkongsamba Health District (NHD), Cameroon.

## Methodology

The methods approved for obtaining the data have been published in PLoS One [16]. They have been summarised below with highlights of the sections used to obtain data for this work.

### Study design and setting

We conducted a hospital-based cross-sectional study targeting women in late pregnancy from four major health facilities of the Nkongsamba Health District (NHD). Contrary to the initial indications on the protocol that data were to be collected only from the NRH, 4 major health facilities of the NHD were included in the study to work with a more representative population. These health facilities included the NRH, Catholic Medicalised Health Centre, Fultang

polyclinic and the Bon Samaritain Medicalised Health Centre. These health facilities are responsible for over 85% of deliveries that occur in the health district.

Data were collected using a semi-structured interview-administered questionnaire and by blood assays. All apparently healthy women were involved in the study. Data collection for this paper was done between November 2020 and September 2021.

## Target population and sampling

Eligible participants were all pregnant women received at the antenatal care unit of the maternity of the four health facilities in late pregnancy (at least at 37 weeks of gestation) [16]. Contrary to the information presented on the protocol, which planned to recruit only apparently healthy women with no potential causes of hypocalcaemia, this project recruited all apparently healthy pregnant women to report a general prevalence of hypocalcaemia. The minimum required sample size stated in the protocol was 1067 participants, and participants were consecutively included [16] in the study as they arrived at the maternity of these health facilities. Recruitment stopped when the required sample size was reached, and non-response cases were replaced.

## Procedure of implementation and data collection

The research questionnaire was pretested at the Kekem District Hospital, West of Cameroon, while administrative authorisations were obtained from the District Medical Officer of the NHD and the directors of the respective health facilities. Data collection was done under supervision by seven midwives working in the included maternities. These data collectors were trained in three training sessions, which lasted five hours each on the consenting and data collection procedures.

Only eligible and consenting pregnant women were allowed to participate in this study. Questionnaires were administered face-to-face, on a one-to-one basis, and data gathered included sociodemographic, economic, and obstetric characteristics. In addition to the interview, 10ml of total blood were collected into lithium heparinised tubes following the WHO best practices in phlebotomy using vacuated needles on unsqueezed veins to measure total calcaemia. These samples were immediately centrifuged and used to measure total calcaemia and albuminaemia. Moreover, 10ml of blood was collected into dry vacutainer tubes (without anticoagulant) and settled for 30 minutes for serum extraction (immediately used for ionised calcium and pH assays). All blood samples were collected on seated participants after breathing calmly for 10 minutes, with no prior physical exercise. During this procedure, tourniquets were applied only for less than 1 minute, arm exercise and fist formation were discouraged [16].

Payne's equation [17, 18] is a valid correction equation when albumin concentrations are measured using Bromocresol Green (BCG) [19]. However, at very low albumin concentrations, normocalcaemic cases risk to be classified to be hypercalcaemic [20]. According to a recent analysis of existing evidence, total calcium levels should be used instead of the albumin-corrected calcium in patients with no suspected disorder of calcium homeostasis [21]. Given the recent controversy on this subject and no established consensus in pregnancy, our findings were presented using the total and albumin-corrected calcium concentrations.

Total blood calcium and albumin were measured by atomic absorption spectrophotometry (AAS) using the semi-automatic KENZA MAX spectrophotometer manufactured by BIOLABO. Total blood calcium and albumin were measured using the O-Cresol Phtalein Complexone (CPC) and bromocresol green (BCG) methods, respectively (as per the BIOLABO standard operating procedures) [22, 23]. Crude ionised calcium concentrations and pH were

measured by ion-selective electrode potentiometry (ISEP) using the K-lite 8 (MSLEA15-H model) serum electrolyte analyser and its standard reagents following standard manufacturer protocols [3]. Crude ionised calcium concentrations were then corrected for any pH variations around 7.4 using the equation Corrected iCa$^{2+}$ (pH 7.4) = Measured iCa$^{2+}$ [1–0.53 × (7.40– measured pH)] [24] which is valid between 7.2–7.6.

## Data analysis

A data entry sheet was developed on Epi-Info version 7.2.4.0, and all validated questionnaires were entered into the sheet. Statistical analysis was carried out using Epi-Info version 7.2.4.0. Questionnaires were evaluated for completeness and the absence of key information like the albuminemia, crude ionised and total calcaemia. When vital information key to the study was missing, the questionnaire was rejected. Major frequencies or proportions were estimated with their 95% confidence intervals (CIs) for categorical variables (prevalence of hypocalcaemia, marital status, religion, and occupation), while the mean ($\mu$) and standard deviation (SD) were used for continuous variables (age, calcaemia, and albuminemia). Participants with total crude calcium concentrations or total albumin-corrected calcium concentrations less than or equal to 85mg/L [3, 6] were considered to be hypocalcaemic. In the case of ionised hypocalcaemia, a cut-off of less than or equal to 45mg/L was used [25]. All women who had ionised, total crude, and albumin-corrected total calcium concentrations less than or equal to the stated cut-offs were classified as hypocalcaemic (low blood calcium).

Concerning logistic regression, total crude hypocalcaemia (total calcaemia measured by AAS and hypocalcaemia defined with a cut-off of 85mg/L) and ionised hypocalcaemia (ionised calcaemia measured by ISEP, and hypocalcaemia defined with a cut-off of 45mg/L) were used as the outcomes of interest. Because of the absence of identical studies, most of the categorisation of sociodemographic characteristics for logistic regression was done following directions of effect [3, 26] and examples from previous studies among pregnant women [27]. Other variables like the marital status were categorised following standard divisions like in union/out of union [3]. Obstetric factor categorisation was done based on evidence from other studies [28–30] and WHO indicators antenatal care coverage [31, 32]. People who consume fruits rarely are more exposed to hypocalcaemia than their counterparts who take fruits more often [14]. So, women who took fruits rarely (less than 2 times a week) were compared with those who took fruits more often. The strength of association between hypocalcaemia (ionised and total) and selected covariates like monthly revenue, level of education, number of antenatal visits, and nutritional habits was measured using the odds ratio (OR), and 95% CI generated during logistic regression. Variables with $p$-values $< 0.25$ were considered for multiple logistic regression. In a multiple logistic regression model, each eligible variable from simple logistic regression was controlled for all other eligible variables. All the variables eligible for multiple logistic regression remained in the model with no backward elimination. The threshold of significance for this analysis was set at a $p$-value of 0.05.

## Ethics statement

Ethical clearance was obtained from Cameroon Bioethics Initiative (CAMBIN)/ Ethics Review and Consultancy Committee (ERCC). The ethical clearance reference number is CBI/452/ ERCC/CAMBIN. Only consenting participants who signed informed consent forms (or assent forms for minors) were included in this study. The confidentiality and autonomy of all participants were respected, and they were free to leave the study at any time.

**Table 1. Sociodemographic characteristics of participants.**

| Characteristic | Modalities | Frequency | Proportions (%) |
|---|---|---|---|
| Age groups (*n* = 1074) | 15–20 years | 97 | 09.03 |
| | 21–30 years | 607 | 56.52 |
| | 31–50 years | 370 | 34.45 |
| Marital status (*n* = 1068) | Single | 371 | 34.74 |
| | Married | 269 | 25.19 |
| | Cohabiting | 462 | 39.89 |
| | Widow | 02 | 0.19 |
| Level of education (*n* = 1074) | Primary | 72 | 06.60 |
| | Secondary | 741 | 68.99 |
| | Higher | 261 | 24.30 |
| Monthly revenue in thousand FCFA (*n* = 1059) | < 50 (USD 89) | 119 | 11.24 |
| | 50–100 (USD 89–179) | 270 | 25.50 |
| | 100–200 (USD 179–358) | 456 | 43.06 |
| | > 200 (USD 158) | 214 | 20.21 |
| Religion (*n* = 1056) | Atheist | 64 | 06.06 |
| | Catholic | 518 | 49.05 |
| | Muslim | 33 | 03.13 |
| | Protestant | 441 | 41.76 |
| Number of household occupants (*n* = 1053) | 1–4 | 438 | 41.60 |
| | 5–7 | 415 | 39.41 |
| | $\geq$ 7 | 200 | 18.99 |

## Results

### Characteristics of the population

This study included a total of 1074 eligible individuals with a non-response rate of 6.61%. The mean age of the participants was 28.20±6.08 years, with the 21–30 years age group being the most represented (56.52%). Table 1 shows the sociodemographic characteristics of the participants. About 9 in 10 (93%) had attended secondary school, and 65% were in a union. Concerning religion and monthly revenue, 90.81% were Christians and 11.24% earned below 50.000FCFA (89 USD) every month.

The obstetric characteristics of the participants are presented in Table 2. The mean number of pregnancies and the average gestational age at the first antenatal visit (for the current pregnancy) was 3.46±2.06 and 17.10±8.08 weeks, respectively. About 2 in 10 women were in their first pregnancy, while a majority of the participants were in their second, third or fourth pregnancy (47.06%). About half (53.52%) initiated antenatal care after 4–6 months of pregnancy, while only about 1 in 10 women (11.62%) started antenatal care within the first two months of pregnancy. Concerning antenatal visits, 87.14% had attended at least 4 antenatal care sessions.

### Prevalence of hypocalcaemia in late pregnancy

In this study, the prevalence of total crude and albumin-corrected hypocalcaemia were 61.64 [58.69–64.50]% and 56.70 [53.72–59.64]%, respectively. The difference in the above two proportions was statistically significant with a *p*-value of 0.000. From the pH-corrected ionised calcium point of view, the prevalence of ionised hypocalcaemia was 2.89 [2.04–4.07]%. When compared with the total crude and total albumin-corrected prevalence of hypocalcaemia, this showed statistically significant differences with *p*-values of 0.000 in each case.

**Table 2. Obstetric characteristics of participants.**

| Characteristic | Modalities | Frequency | Proportion (%) |
|---|---|---|---|
| Gestational age at first antenatal visit (n = 1050) | Before 2 months | 122 | 11.62 |
| | 2–3 months | 263 | 25.05 |
| | 4–6 months | 562 | 53.52 |
| | Above 7 months | 103 | 09.81 |
| Number of antenatal visits (n = 1034) | No antenatal care | 09 | 0.87 |
| | 1–3 | 124 | 11.99 |
| | 4–6 | 554 | 53.58 |
| | > 6 | 347 | 33.56 |
| Total number of pregnancies (n = 1071) | First pregnancy | 256 | 23.90 |
| | 2–4 | 504 | 47.06 |
| | 5–8 | 289 | 26.98 |
| | > 8 | 22 | 02.05 |
| Total number of deliveries (n = 1067) | None | 249 | 23.34 |
| | 1–3 | 596 | 55.86 |
| | 4–6 | 205 | 19.21 |
| | > 6 | 17 | 01.59 |

Only 28 participants (2.61 [1.81–3.74]%) had hypocalcaemia when defined using the three methods (ionised, total crude and albumin corrected total calcium) at the same time. Fig 1 shows the scatter plot of the relationship between ionised calcaemia and total crude calcaemia, while in Fig 2, ionised calcaemia is plotted against albumin-corrected total calcium. The two

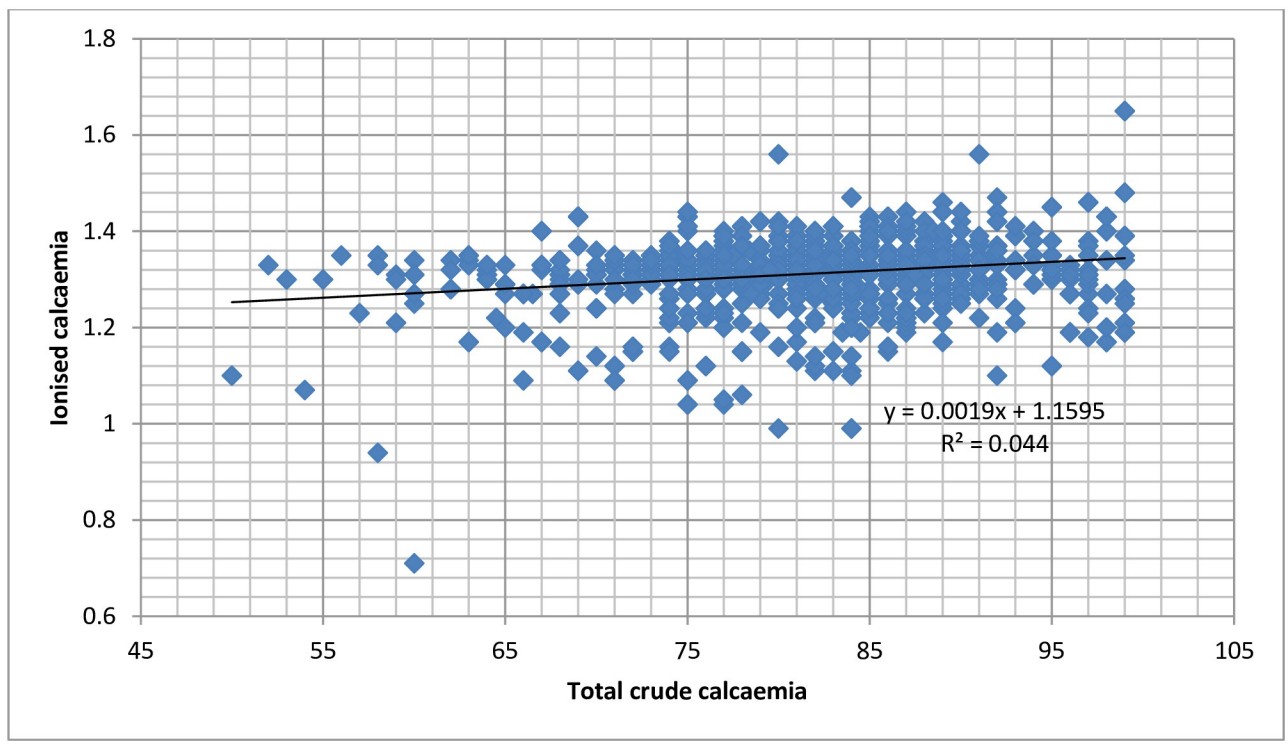

y = 0.0019x + 1.1595
R² = 0.044

**Fig 1. Plot of the relationship between ionised calcaemia and total crude calcaemia in pregnancy.**

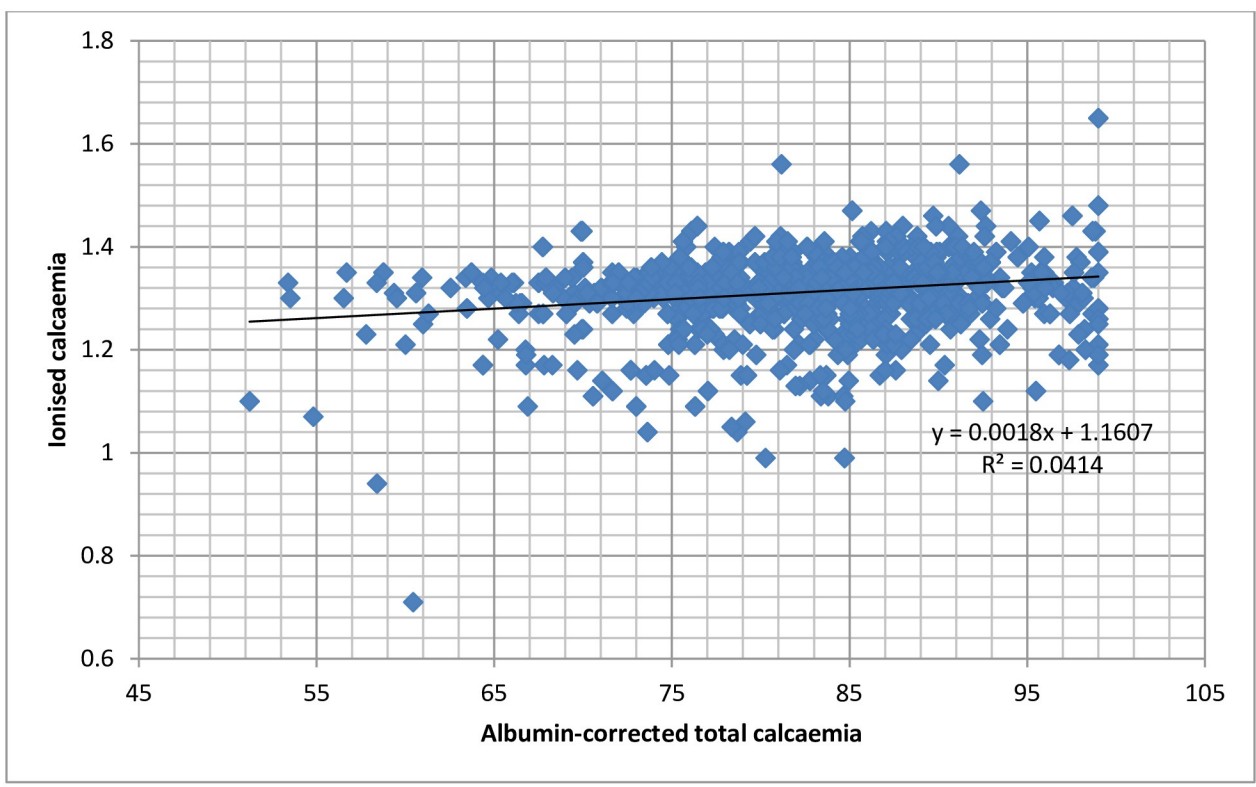

**Fig 2. Plot of the relationship between ionised calcaemia and albumin-corrected total calcaemia in pregnancy.**

plots show that trivial increases in ionised calcaemia are associated with more considerable changes in total crude and albumin-corrected calcaemia. The goodness-of-fit measure ($R^2$) in both plots shows that the strength of the relationship is very weak, with $R^2$ less than 5% in both cases. The probability of having the gradient zero in each case was 0.000.

The mean concentrations of total albumin-corrected and total crude calcaemia (uncorrected calcaemia) were 82.86±8.89 mg/L and 82.20±8.84 mg/L, respectively. The mean crude and pH-corrected ionised calcaemia among these 1074 participants were 52.87±3.83 mg/L and 52.61±3.07 mg/L, respectively. Women classified to have total crude hypocalcaemia had their odds of ionised hypocalcaemia increased by 5.88 folds compared with their counterparts with total normocalcaemia (*p*-value = 0.003).

## Factors associated with total crude hypocalcaemia in pregnancy

Sociodemographic and obstetric factors associated with total hypocalcaemia are presented in Table 3. Following simple logistic regression, only monthly revenue and religion were significantly associated with hypocalcaemia. When controlled in a multiple logistic regression model with six eligible factors (*p*-value < 0.25), only monthly revenue showed significant association with total crude hypocalcaemia. Participants who earned less than 100.000FCFA (179 USD) in a month had 0.73 lower odds to have total crude hypocalcaemia than those who earned more. No factors related to antenatal care were found to be statistically significant.

Table 4 presents nutritional factors associated with total crude hypocalcaemia. Following simple logistic regression, taking more than two meals a day in pregnancy, taking of desserts, cheese consumption, and consumption of cereals like rice and corn (as the most common

**Table 3. Sociodemographic and obstetric factors associated with total crude hypocalcaemia.**

| Factor | Simple logistic regression | | Multiple logistic regression | |
|---|---|---|---|---|
| | OR [95% CI] | p-value | AOR [95% CI] | p-value |
| Age below 30 (Y/N) | 0.93 [0.72–1.21] | 0.604 | | |
| Level of education of participant above secondary (Y/N) | 0.86 [0.65–1.15] | 0.315 | | |
| Level of education of partner above secondary (Y/N) | 0.79 [0.60–1.03] | 0.079 | 0.81 [0.60–1.09] | 0.163 |
| Less than 7 household occupants (Y/N) | 1.19 [0.91–1.56] | 0.198 | 1.13 [0.85–1.51] | 0.394 |
| Marital status as single (Y/N) | 0.90 [0.69–1.16] | 0.416 | | |
| Gestational age at first visit below 4 months (Y/N) | 1.05 [0.81–1.36] | 0.707 | | |
| Number of ANC visits above 3 (Y/N) | 0.97 [0.67–1.36] | 0.878 | | |
| Occupation other than housewife (Y/N) | 0.80 [0.62–1.04] | 0.101 | 0.94 [0.70–1.25] | 0.669 |
| Monthly revenue below 100.000 FCFA (Y/N) | 0.75 [0.58–0.97] | 0.027 | 0.73 [0.55–0.96] | 0.024 |
| Non-Catholics (Y/N) | 1.30 [1.01–1.67] | 0.039 | 1.23 [0.95–1.59] | 0.120 |
| Always or sometimes reminded or supported by partner to take supplements (Y/N) | 0.79 [0.56–1.11] | 0.183 | 0.74 [0.52–1.07] | 0.115 |

- *p*-value < 0.05: Statistical significance,
- *p*-value < 0.25: Eligible for multiple regression analysis,
- All the variables eligible for multiple logistic regression remained in the model with no backward elimination,
- Total hypocalcaemia: Total crude calcium concentration less than or equal to 85.0 mg/L (Measured by atomic AAS using the O-Cresol Phtalein Complexone reagent),
- Prevalence of total crude hypocalcaemia (662/1074) = 61.64%,
- Y/N = Yes/No, OR = Odds Ratio, AOR = Adjusted Odds Ratio, CI = Confidence Interval.

**Table 4. Nutritional factors associated with total crude hypocalcaemia.**

| Factor | Simple Logistic regression | | Multiple logistic regression | |
|---|---|---|---|---|
| | OR [95% CI] | p-value | AOR [95% CI] | p-value |
| Daily number of meals above 2 (Y/N) | 0.68 [0.52–0.88] | 0.004 | 0.68 [0.52–0.90] | 0.006 |
| Taking desserts between meals (Y/N) | 0.74 [0.55–0.99] | 0.043 | 0.73 [0.54–0.99] | 0.046 |
| Consumption of plantain, banana and tubers compared with Others (like corn and rice) as most common content of the meal | 1.36 [1.06–1.75] | 0.015 | 1.37 [1.08–1.80] | 0.012 |
| Taking fruits rarely (Y/N) | 0.82 [0.50–1.34] | 0.434 | | |
| Taking oranges, watermelon, and pineapples as the most consumed fruit (Y/N) | 1.06 [0.81–1.37] | 0.689 | | |
| Never taken cheese (Y/N) | 1.25 [0.97–1.60] | 0.080 | 1.11 [0.85–1.45] | 0.430 |

- *p*-value < 0.05: Statistical significance,
- *p*-value < 0.25: Eligible for multiple regression analysis,
- All the variables eligible for multiple logistic regression remained in the model with no backward elimination,
- Total hypocalcaemia: Total crude calcium concentrations less than or equal to 85.0 mg/L (Measured by AAS using the O-Cresol Phtalein Complexone reagent),
- Prevalence of total crude hypocalcaemia (662/1074) = 61.64%,
- Y/N = Yes/No, OR = Odds Ratio, AOR = Adjusted Odds Ratio, CI = Confidence Interval.

**Table 5. Sociodemographic and obstetric factors associated with ionised hypocalcaemia.**

| Factor | Simple logistic regression | | Multiple logistic regression | |
|---|---|---|---|---|
| | OR [95% CI] | *p*-value | AOR [95% CI] | *p*-value |
| Age below 30 (Y/N) | 0.55 [0.27–1.13] | 0.102 | 0.44 [0.20–0.96] | 0.039 |
| Level of education of participant above secondary (Y/N) | 1.74 [0.82–3.69] | 0.146 | 3.27 [1.23–8.70] | 0.017 |
| Level of education of partner above secondary (Y/N) | 1.33 [0.53–2.43] | 0.749 | | |
| Less than 7 household occupants (Y/N) | 0.92 [0.43–1.97] | 0.828 | | |
| Marital status as single (Y/N) | 1.90 [0.92–3.92] | 0.084 | 2.54 [1.15–5.62] | 0.021 |
| Gestational age at first visit below 4 months (Y/N) | 2.01 [0.97–4.18] | 0.059 | 2.47 [1.10–5.56] | 0.029 |
| Number of ANC visits above 3 (Y/N) | 0.55 [0.22–1.39] | 0.207 | 0.38 [0.13–1.11] | 0.076 |
| Occupation other than housewife (Y/N) | 0.62 [0.30–1.26] | 0.186 | 0.34 [0.13–0.89] | 0.027 |
| Monthly revenue below 100.000 FCFA (Y/N) | 1.09 [0.52–2.27] | 0.817 | | |
| Non-Catholics (Y/N) | 1.46 [0.70–3.06] | 0.317 | | |
| Always or sometimes reminded or supported by partner to take supplements (Y/N) | 0.69 [0.27–1.72] | 0.422 | | |

• *p*-value < 0.05: Statistical significance,

• *p*-value < 0.25: Eligible for multiple regression analysis,

• All the variables eligible for multiple logistic regression remained in the model with no backward elimination,

• Ionised hypocalcaemia: pH-corrected ionised calcium concentration less than or equal to 45.0 mg/L (Measured by ISEP),

• Prevalence of ionised hypocalcaemia (31/1074) = 2.89%,

• Y/N = Yes/No, OR = Odds Ratio, AOR = Adjusted Odds Ratio, CI = Confidence Interval.

food) were significantly associated with lower odds of total crude hypocalcaemia in pregnancy. When controlled for confounders in a multiple logistic regression model, women who took more than 2 meals a day (AOR = 0.68 [0.52–0.90], *p*-value = 0.006) and those who took desserts (AOR = 0.73 [0.54–0.99], *p*-value = 0.046) had their odds of total crude hypocalcaemia reduced by 0.68 and 0.74 fold, respectively. Moreover, having banana/plantain and tubers as the content of their most consumed meal (as opposed to cereals) significantly increased the odds of total crude hypocalcaemia by 1.37 folds (AOR = 1.37 [1.08–1.80], *p*-value = 0.012).

Table 5 presents the sociodemographic and obstetric factors associated with ionised hypocalcaemia. Contrary to total hypocalcaemia, which had monthly revenue as the only sociodemographic factor, 5 factors were found to have a significant association with ionised hypocalcaemia. Age, level of education, marital status, gestational age at the first visit, and the participant's occupation had a statistically significant association with ionised hypocalcaemia. Women below 30 years had 0.44 lower odds of ionised hypocalcaemia compared with their older counterparts (AOR = 0.44 [0.20–0.96], *p*-value = 0.039). Those with higher education had 3.27 higher odds of ionised hypocalcaemia compared with women with lower education (AOR = 3.27 [1.23–8.70], *p*-value = 0.017). Single women had their odds for ionised hypocalcaemia increased by 2.54 folds compared with other women in union (AOR = 2.54 [1.15–5.62], *p*-value = 0.021). Women who started ANC before 4 months of pregnancy had their odds of ionised hypocalcaemia increased by 2.47 folds compared with their counterparts who started later (AOR = 2.47 [1.10–5.56], *p*-value = 0.029). Moreover, women with occupations other than housewife had their odds of ionised hypocalcaemia reduced by 0.34 folds (AOR = 0.34 [0.13–0.89], *p*-value = 0.027).

Table 6 presents nutritional factors associated with ionised hypocalcaemia (only one variable was eligible for multiple logistic regression). Contrary to total hypocalcaemia, which was

**Table 6. Nutritional factors associated with ionised hypocalcaemia in pregnancy.**

| Factors | Simple Logistic regression | |
|---|---|---|
| | OR [95% CI] | *p*-value |
| Daily number of meals above 2 (Y/N) | 0.71 [0.34–1.45] | 0.344 |
| Taking desserts between meals (Y/N) | 0.45 [0.21–0.94] | 0.034 |
| Consumption of plantain, banana and tubers compared with Others (like corn and rice) as most common content of the meal | 1.08 [0.52–2.25] | 0.839 |
| Taking fruits rarely (Y/N) | 1.66 [0.48–5.69] | 0.416 |
| Taking oranges, watermelon, and pineapples as the most consumed fruit (Y/N) | 0.72 [0.32–1.62] | 0.420 |
| Never taken cheese (Y/N) | 1.48 [0.70–3.13] | 0.299 |

• *p*-value < 0.05: Statistical significance

• Ionised hypocalcaemia: pH-corrected ionised calcium concentration less than or equal to 45.0 mg/L (Measured by ISEP).

• Prevalence of ionised hypocalcaemia (31/1074) = 2.89%,

• Y/N = Yes/No, OR = Odds Ratio, CI = Confidence Interval.

more associated with nutritional factors, ionised hypocalcaemia was associated only with the habit of taking desserts between meals. Women who took desserts between meals had their odds of ionised hypocalcaemia reduced by 0.45 folds (OR = 0.45 [0.21–0.94], *p*-value = 0.034).

Table 7 evaluates calcium supplementation as a factor influencing calcaemic states. From the table, calcium supplemented women had respectively, 0.84 and 0.79 lower odds of total (*p*-value = 0.231) and ionised hypocalcaemia (*p*-value = 0.538) compared with non-supplemented women. Also, taking calcium supplements at the same time as other supplements (iron and folic acid) significantly reduced the odds of total crude hypocalcaemia in pregnancy (OR = 0.69 [0.50–0.96], *p*-value = 0.027). Taking more than 1000mg of elemental calcium a day was associated with reduced odds of total crude hypocalcaemia (OR = 0.75 [0.55–1.01], *p*-value = 0.061). No statistically significant association was found between ionised hypocalcaemia and calcium supplementation variables.

**Table 7. Logistic regression analysis between calcium supplementation variables and likelihood of total crude and ionised hypocalcaemia.**

| Calcium supplementation variable | Total crude hypocalcaemia | | Ionised hypocalcaemia | |
|---|---|---|---|---|
| | OR [95% CI] | *p*-value | OR [95%CI] | *p*-value |
| Calcium supplementation (Y/N) | 0.84 [0.63–1.12] | 0.231 | 0.79 [037–1.69] | 0.538 |
| Duration of calcium supplementation below 4 months (Y/N) | 0.89 [0.62–1.27] | 0.522 | 0.40 [0.09–1.74] | 0.224 |
| Taking calcium supplements with other supplements (Y/N) | 0.69 [0.50–0.96] | 0.027 | 0.80 [0.32–2.03] | 0.651 |
| Daily dose of elemental calcium supplementation above 1000mg (Y/N) | 0.75 [0.55–1.01] | 0.061 | 2.32 [0.92–5.82] | 0.073 |

• *p*-value < 0.05: Statistically significance,

• Total hypocalcaemia: Total crude calcium concentrations less than or equal to 85.0 mg/L (Measured by AAS using the O-Cresol Phtalein Complexone reagent),

• Ionised hypocalcaemia: pH-corrected ionised calcium concentration less than or equal to 45.0 mg/L (Measured by ISEP).

• Prevalence of total crude hypocalcaemia (662/1074) = 61.64%,

• Prevalence of ionised hypocalcaemia (31/1074) = 2.89%,

• Y/N = Yes/No, OR = Odds Ratio, CI = Confidence Interval.

## Discussion

The main goal of our study was to estimate the prevalence of hypocalcaemia in apparently healthy pregnant women in the third trimester from the ionised and total calcaemia viewpoints. In this work, factors associated with ionised and total hypocalcaemia were also evaluated. The prevalence of total crude and total albumin-corrected hypocalcaemia in this study were 61.64 [58.69–64.50]% and 56.70 [53.72–59.64]%, respectively (p-value = 0.000), while the prevalence of ionised hypocalcaemia was as low as 2.89 [2.04–4.07]%.

It has been argued that total crude and albumin-corrected calcaemia in pregnancy have shortcomings in defining the exact physiological calcaemic states. Defining hypocalcaemia in pregnancy from the viewpoint of total calcaemia exaggerates the burden of this condition. The enormous discrepancy observed between the results of hypocalcaemia in pregnancy when defined from the total and ionised calcium point of view has been described in literature. The common laboratory finding of low circulating total calcium in pregnancy is far from directly defining the physiologically active concentration and is more of a factitious hypocalcaemia [6].

This discordance between total and ionised calcaemia in classifying calcaemic states has been described in other studies. In a study on outpatients suspected of bone or calcium metabolism disorders, there was 92% disagreement between ionised and total calcaemia in classifying hypocalcaemic states [33].

The scientific evidence suggesting that ionised calcium concentrations remain constant throughout pregnancy despite the variations in total calcium concentrations [6, 34] has limited exploration of the burden of ionised hypocalcaemia in pregnant women. Even if the state of pregnancy does not affect ionised calcaemic states, there are many potential causes of hypocalcaemia that can exist in association with pregnancy. These causes are dominated by vitamin D deficiencies, hypoparathyroidism, nutritional deficiencies and malnutrition, and some chronic diseases [6]. Therefore, ionised hypocalcaemia can be a problem in some pregnancies. To the best of our knowledge, no study has determined the prevalence of ionised hypocalcaemia in pregnancy.

This state of ionised hypocalcaemia is directly associated with adverse effects on maternal calcium-related physiology and the growth of the foetus [6]. Calcium has been identified to interfere in practically all physiological activities in the human body. It is indispensable as a second messenger in major signalling pathways and contributes to regulating cell excitability, exocytosis, apoptosis, and cell movement. Even though known to be governed mainly by the phospholipase C pathway, $Ca^{2+}$ signalling can be engaged by many other pathways, including pathways for cell growth, differentiation and cell death. Calcium and calcium signalling are indispensable for vital activities like cell death, muscle contraction, neuronal transmission, neurogenesis, gene transcription, exocytosis, cell movement, cell growth and proliferation, synaptic plasticity, enzyme activity, secretion of saliva, and more [35, 36]. Low levels of ionised calcaemia in pregnancy are likely to interfere with most of these processes, foetal growth and differentiation-related activities.

The high prevalence of total hypocalcaemia and the cases of ionised hypocalcaemia in pregnancy in this population might be due to the high prevalence of vitamin D deficiency and insufficiency that have been reported in low and middle-income countries. A study in Buea (Cameroon) reported a prevalence of vitamin D deficiency and insufficiency in adults beyond 35 years of 3.2% and 22.6% [37]. In neighbouring Nigeria, the third-trimester prevalence of vitamin D deficiency and insufficiency of 22.5% and 60%, respectively, have been reported [38]. The prevalence of hypoparathyroidism in the general population is low, ranging from 0.5–6.6% [39], while the prevalence in pregnancy has not been reported [40].

The recorded prevalence of ionised hypocalcaemia in this study is likely due to subclinical hypovitaminosis D, hypoparathyroidism or other subclinical causes of hypocalcaemia in pregnancy. Therefore, apparently healthy pregnant women are likely to have potentially life-threatening pathologies liable to cause a fall in ionised calcaemia. In our study, this occurs in about 3 of every 100 pregnant women. This is not negligible, given that it could be associated with severe adverse maternofoetal morbidities. Pregnant women presenting with symptoms of hypocalcaemia should not only be placed on calcium therapy, but a complete exploration into the specific cause in pregnancy needs to be carried out.

As reported by studies in India, Algeria, and Cameroon, the prevalence of total hypocalcaemia in pregnancy is very high. These studies evaluated the prevalence of hypocalcaemia in the third trimester and found 66%, 70%, and 59% in India, Algeria, and Cameroon, respectively. The study in Cameroon was carried out in one of the health facilities considered in this study (NRH) and reported a 95% confidence interval of 53.42–63.90%, which perfectly intersects the confidence interval of the current research (56.70 [53.72–59.64]%).

According to the findings of this new study, at least 5 in 10 pregnant women in the NHD have low calcium concentrations in the third trimester. When total crude calcium concentrations are considered, the prevalence of hypocalcaemia increases slightly but significantly to 61.64 [58.69–64.50]% in this population. The slight discrepancy between these two results from Cameroon can be explained by an extension of the study population to include women from three other major health facilities in the NHD. Also, the prevalence of hypocalcaemia reported in India and Algeria are slightly higher than findings in Cameroon but similar to the prevalence recorded in this study with total crude (unadjusted or uncorrected) calcium concentrations (61.64%). These studies considered lower cut-offs (less than 80mg/L) to define hypocalcaemia, which might give us a lower prevalence if considered in this study. Moreover, Cameroon's nutritional habits and sociodemographic characteristics are very different from those in India and Algeria (a possible explanation for the observed discrepancies).

However, a study in Maiduguri, Nigeria, reported a prevalence of hypocalcaemia in pregnancy of 29% (about half the prevalence in the present study) [4]. The inclusion criteria in this study could explain the low prevalence, given that they included pregnant women of all gestational ages [4]. Given that calcium absorption in pregnancy doubles within the first 12 weeks [34], some participants must have been recruited in the first and second trimesters when calcium concentrations are relatively high and foetal demands still low.

Should total hypocalcaemia in pregnancy be ignored, as suggested by some authors? Our study found out that total crude hypocalcaemia was significantly associated with increased odds for ionised hypocalcaemia. Women classified as having total crude hypocalcaemia had their odds of ionised hypocalcaemia increased by 5.88 folds compared with their counterparts with total normocalcaemia (*p*-value = 0.003). This significant association still brings to light the possibility of predicting ionised calcaemic states from total calcaemia. However, these equations are generally complex given that ionised calcium concentrations depend on several variables like the pH, total albumin, plasma proteins, and concentration of some ions in the blood. Even though total hypocalcaemia in pregnancy might exaggerate the burden, it is a probable predictor of the likelihood of ionised hypocalcaemia in these women. If women with total hypocalcaemia have their odds of ionised hypocalcaemia increased by about 6 folds, women diagnosed with total hypocalcaemia should be systematically checked for variations of the metabolically active fraction.

Apart from a monthly revenue below 100.000FCFA (179 USD), no sociodemographic factor had a significant association with total crude hypocalcaemia. Notwithstanding, a study carried out in Maiduguri teaching hospital (Nigeria) reported the level of education to be significantly associated with a reduced likelihood of total crude hypocalcaemia [4]. A similar

trend was observed in this study but was not statistically significant. Our findings on the relationship between age and total hypocalcaemia also concord with results from Nigeria [4].

We found no studies evaluating the influence of monthly income on the likelihood of total hypocalcaemia in pregnancy. Our findings suggest that poorer women were less likely to have total crude hypocalcaemia than their rich counterparts. This relationship is complex given that from expectations, rich women should feed well and have money to buy calcium supplements than poor women. However, this effect could be associated with increased access of poor women to locally cultivated calcium-containing foodstuffs than the richer women who might have to buy everything from the market.

Notwithstanding, nutritional variables showed statistically significant associations with total crude hypocalcaemia. Women who took at least 3 meals a day had significantly reduced odds of total crude hypocalcaemia. This is logical as adequate nutritional intake is associated with an increased likelihood of meeting calcium demands. Moreover, taking desserts between meals reduced the odds of total crude hypocalcaemia by 0.73 folds. Desserts are usually dairy products or derivatives of dairy products and constitute any other food of fruits taken between meals. This additional intake of food/fruit can potentially lead to relatively more calcium intake a day. We found no studies that associated desserts consumption with low serum calcium in pregnancy.

Furthermore, consumption of tubers, plantains/bananas as opposed to cereals like rice and corn was associated with increased odds of total crude hypocalcaemia. Consumption of potassium-rich diets could indirectly induce pseudo-hypoparathyroidism and cause hypocalcaemia in pregnancy [41]. This could have been the case with consumption of meals with potassium-rich content like plantains and bananas [42]. According to a study carried out in Nkongsamba on the calcium content of some commonly consumed meals, cereal-containing meals were found to have relatively high calcium content compared with meals made from tubers, plantain and bananas [43]. However, in the same study, these meals contained relatively high amounts of phytates and moderate concentrations of oxalates which inhibit calcium absorption [43]. Studies targeting related nutritional matters need to be designed in this setting to evaluate the mineral content (particularly calcium) and relative concentrations of calcium absorption inhibitors (as well as the *invitro* calcium availability) of these tubers and cereals in both cooked and uncooked states.

Hence, this study highlights through these factors that adequate dietary intake is indispensable in meeting calcium needs in pregnancy. Pregnant women should be encouraged to take at least three meals a day, take desserts and consume more cereals to maximise the chances of averting hypocalcaemia. These recommendations align with WHO recommendations on nutrition in pregnancy [9].

From the viewpoint of ionised hypocalcaemia, women above 30 years, women with higher education, single women, women who initiated ANC before 4 months of pregnancy, housewives and women who did not take desserts between meals had their odds of ionised hypocalcaemia significantly increased. No research has evaluated the sociodemographic, obstetric and nutritional factors affecting ionised hypocalcaemia. Our findings suggest that even though ionised calcaemic states depend on physiological changes and total calcaemia, upstream factors (such as sociodemographic, obstetric and nutritional factors) may have a crucial role (direct or indirect) to play by influencing the physiological state of these women. More research is required to better elucidate the role played by these upstream factors on ionised calcaemic states, especially in pregnancy.

Although not statistically significant, calcium supplementation reduced the odds of having all forms of hypocalcaemia in pregnancy. The observed effect is clinically significant given that it is beneficial, acceptable, cost-effective and easy to implement despite its statistical

insignificance [44]. This relationship between calcium supplementation and calcaemic states is at the heart of the WHO recommendations on calcium supplementation among women with insufficient dietary intake [9, 45, 46].

Taking calcium supplements together with other supplements was associated with reduced odds of total crude hypocalcaemia. As expected with ionised hypocalcaemia, the strength of association was weaker. This relationship could be explained from the viewpoint of adherence. Participants who took all their supplements at the same time are likely to have been more adherent to calcium supplementation, given that adherence to iron and folic acid supplements is higher among pregnant women [47–49]. Taking more than 1000mg of elemental calcium had a negative non-statistically significant association with total hypocalcaemia. Nevertheless, the association carried some tendency of statistical significance with a *p*-value of 0.061. This observation aligns with expectations and highlights the necessity of high dose supplementation suggested by the WHO. The tendency of statistical significance might be due to the sample size, but the strength of association is relatively considerable (OR = 0.75).

Surprisingly, women who took more than 1000mg of elemental calcium a day had their odds of ionised hypocalcaemia increased by 2.32 folds. This unexpected result contrasts with the findings on total crude hypocalcaemia given that, amongst others, ionised calcaemia depends on total calcaemia. This observation could be due to the small proportion of women with ionised hypocalcaemia and thus, warrants a more extensive exploration.

The results presented in this write-up should be exploited with full knowledge of the limits. We presented the prevalence for total crude, albumin-corrected and ionised hypocalcaemia in late pregnancy but failed to hormonally explore women with ionised hypocalcaemia to find full explanations. The physiological implications and consequences of hypocalcaemia were not addressed in our study. Some of the variables adopted as nutritional factors were not quantifiable and could not be applied directly. For example, fruit consumption was not evaluated in terms of type and number of pieces a day (which was very difficult to evaluate in our context). Furthermore, we did not make any difference between chronic hypocalcaemia and acute changes in calcaemic states. The cross-sectional design adopted in this research could allow for the detection of associations that are not permanent and hence, subject to variations. Moreover, no causation can be established using our design. However, the design remains the most practical design adapted for exploring such relationships.

## Conclusion and recommendations

Ionised hypocalcaemia in pregnancy is a rare finding. Only 2.89% of all apparently healthy pregnant women have ionised hypocalcaemia in late pregnancy. However, the prevalence of total hypocalcaemia in late pregnancy among apparently healthy women in NHD is as high as 61.64% and 56.70% depending on whether total crude or total albumin-corrected calcium concentrations are considered, respectively. Monthly income below 100.000FCFA (USD 179), taking plantain/banana and tubers as the content of their most consumed meal (opposed to cereals), significantly increase the odds of total hypocalcaemia. Moreover, taking more than 2 meals a day and desserts between meals significantly reduce the odds of total hypocalcaemia in pregnancy.

On the other hand, factors like age above 30 years, having a higher education, being single, having initiated ANC before 4 months of pregnancy, being a housewife and not taking desserts between meals have significantly positive associations with ionised hypocalcaemia.

Total hypocalcaemia in pregnancy should not be neglected but should warrant measurement of the ionised fraction for better classification. Pregnant women with ionised hypocalcaemia should be thoroughly evaluated, and further hormonal exploration carried out. Even

though calcium supplementation is a solution to meeting calcium needs in pregnancy, the fact that principally nutritional factors influence total hypocalcaemia suggests that increased consumption of locally available calcium-rich meals will help combat hypocalcaemia in pregnancy. Further studies should be carried out to understand the causes of ionised hypocalcaemia registered in pregnancy.

## Supporting information

**S1 Data. Data base of determinants and effects of low serum calcium in pregnancy.** (MDB)

## Acknowledgments

Our sincere gratitude goes out to:

- The Director of the NRH and Bethanie Group of laboratories for their support,

- Fouko Eric Dagobert and collaborators for their assistance in the laboratory,

- Matcha Waffo Lea Patricia for contributing to data entry,

- Midwives who participated in data collection, and

- Pregnant women who consented to participate in this study.

Some changes have occurred on the author list compared with the registered protocol published. Additional authors include Ukaogo Prince Onydinma and Fulbert Nkwele Mangala who have contributed significantly by supervision, validation, review and editing.

## Author Contributions

**Conceptualization:** Atem Bethel Ajong, Bruno Kenfack, Innocent Mbulli Ali, Fulbert Nkwele Mangala, Loai Aljerf, Phelix Bruno Telefo.

**Data curation:** Atem Bethel Ajong, Bruno Kenfack, Innocent Mbulli Ali, Martin Ndinakie Yakum, Loai Aljerf, Phelix Bruno Telefo.

**Formal analysis:** Atem Bethel Ajong, Martin Ndinakie Yakum, Phelix Bruno Telefo.

**Funding acquisition:** Atem Bethel Ajong.

**Investigation:** Atem Bethel Ajong, Bruno Kenfack, Innocent Mbulli Ali, Martin Ndinakie Yakum, Ukaogo Prince Onydinma, Fulbert Nkwele Mangala, Loai Aljerf, Phelix Bruno Telefo.

**Methodology:** Atem Bethel Ajong, Bruno Kenfack, Innocent Mbulli Ali, Martin Ndinakie Yakum, Loai Aljerf, Phelix Bruno Telefo.

**Project administration:** Atem Bethel Ajong, Bruno Kenfack, Innocent Mbulli Ali, Loai Aljerf, Phelix Bruno Telefo.

**Resources:** Atem Bethel Ajong, Martin Ndinakie Yakum, Ukaogo Prince Onydinma, Fulbert Nkwele Mangala, Loai Aljerf, Phelix Bruno Telefo.

**Software:** Atem Bethel Ajong, Innocent Mbulli Ali, Martin Ndinakie Yakum, Ukaogo Prince Onydinma, Fulbert Nkwele Mangala, Phelix Bruno Telefo.

**Supervision:** Atem Bethel Ajong, Bruno Kenfack, Innocent Mbulli Ali, Ukaogo Prince Onydinma, Fulbert Nkwele Mangala, Loai Aljerf, Phelix Bruno Telefo.

**Validation:** Atem Bethel Ajong, Bruno Kenfack, Innocent Mbulli Ali, Martin Ndinakie Yakum, Ukaogo Prince Onydinma, Fulbert Nkwele Mangala, Loai Aljerf, Phelix Bruno Telefo.

**Visualization:** Atem Bethel Ajong, Bruno Kenfack, Martin Ndinakie Yakum, Ukaogo Prince Onydinma, Fulbert Nkwele Mangala, Loai Aljerf, Phelix Bruno Telefo.

**Writing – original draft:** Atem Bethel Ajong.

**Writing – review & editing:** Atem Bethel Ajong, Bruno Kenfack, Innocent Mbulli Ali, Martin Ndinakie Yakum, Ukaogo Prince Onydinma, Fulbert Nkwele Mangala, Loai Aljerf, Phelix Bruno Telefo.

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
