## [Decision Letter · Decision Letter 0]

7 Jan 2022

PONE-D-21-31669Low blood calcium in pregnancy: prevalence and risk factors in a resource-limited settingPLOS ONE

Dear Dr. Ajong,

Thank you for submitting your manuscript to PLOS ONE. After careful consideration, we feel that it has merit but does not fully meet PLOS ONE’s publication criteria as it currently stands. Therefore, we invite you to submit a revised version of the manuscript that addresses the points raised during the review process.

We look forward to receiving your revised manuscript.

Kind regards,

Frank T. Spradley

Academic Editor

PLOS ONE

Journal Requirements: 

Reviewers' comments:

Reviewer's Responses to Questions

**Comments to the Author**

1. Does the manuscript adhere to the experimental procedures and analyses described in the Registered Report Protocol?

If the manuscript reports any deviations from the planned experimental procedures and analyses, those must be reasonable and adequately justified.

Reviewer #1: Yes

2. If the manuscript reports exploratory analyses or experimental procedures not outlined in the original Registered Report Protocol, are these reasonable, justified and methodologically sound?

A Registered Report may include valid exploratory analyses not previously outlined in the Registered Report Protocol, as long as they are described as such.

Reviewer #1: Yes

3. Are the conclusions supported by the data and do they address the research question presented in the Registered Report Protocol?

The manuscript must describe a technically sound piece of scientific research with data that supports the conclusions. The conclusions must be drawn appropriately based on the research question(s) outlined in the Registered Report Protocol and on the data presented.

Reviewer #1: Yes

4. Have the authors made all data underlying the findings in their manuscript fully available?

Reviewer #1: Yes

5. Is the manuscript presented in an intelligible fashion and written in standard English?

Reviewer #1: Yes

6. Review Comments to the Author

Please use the space provided to explain your answers to the questions above. (Please upload your review as an attachment if it exceeds 20,000 characters)

Reviewer #1: General comments

This manuscript explores associations of low blood Ca in pregnant women and various sociodemographic, obstetric or nutritional factors. This cross-sectional study was conducted in 4 different health care settings in Cameroon.

Consider using the term low blood Ca in clinically healthy subject throughout the manuscript, instead of hypocalcemia. Please note that the title reads low blood Ca. Hypocalcemia connotates an abnormality with potentially negative clinical effects.

No reference has been cited to support the choice of low blood Ca/hypocalcemia thresholds.

There is a large disagreement in prevalence of low blood Ca, based on the type of the analytical method of choice. This issue deserves further attention. Please consider the following:

1. Visually evaluate your data by plotting iCa (consider the gold standard) vs total calcium, and iCa vs albumin-corrected total Ca.

2. Evaluate the correlation of iCa with tCa and albumin-corrected tCa.

3. Consider evaluating if there is a bias across lab methods, and if this bias is constant.

The criterion used to stratify the sociodemographic (i.e., age, monthly revenue, household occupants), obstetric (i.e., gestational age at first antenatal visit, number of visits, number of pregnancies, number of deliveries), and nutritional factors (i.e., taking fruit rare) should be clear. Describe the reasoning behind the choice of categories.

Authors do not provide enough background to understand the selected nutritional factors as explanatory variables within the context of Cameroon dietarian preferences. For example, it is unclear why authors decided on oranges, watermelons, and pineapples to classify the most consumed fruit. “Eating fruit rarely” it is not a quantifiable variable and may have different meaning across countries or ethnicities (i.e., number of pieces or servings per day).

Specific Comments

L 1. Acknowledge that the study was conducted in Cameroon.

L164. As indicated above, authors need to support the hypocalcemia thresholds with references (iCa, albumin-corrected tCa, and tCa).

L 164. Report the number of cows identified with low Ca serum when combining result from the 3 methods.

L 202. Please explain in materials how data was analyzed to calculate the prevalence of low calcium in blood.

L 217. Consider rephrasing as “Sociodemographic and obstetric factors associated with hypocalcemia are presented in Table 3.”

L 236. Change “Table 8: Nutritional factors affected” by “Table 4: Nutritional factors associated”.

L 250. The first discussion paragraph is written as results. Follow scientific writing guidelines for discussion, and delete it.

L 250. It is striking the difference in prevalence when considering different lab methods. That should be one of the first points of discussion.

L 260 to 262. Alkalogenic diets (rick in K) may induce pseudohypoparathyroidism. Please see http://doi.org/10.3618/jds.2013-7467

L 265. Replace “borderline significance” by “tendency to”. Define in material and methods tendency, for example p< 0.1

L 280. What was the prevalence with unadjusted Calcium levels?

L 296 to 299. Disagreement in Ca concentration across lab methods has been reported in patients with blood Ca metabolism disorders. However, the present study included healthy patients. Are there other plausible explanations? As indicated above, please explore the data.

L 306. Replace regnancies with pregnancies.

L 320. This section of the discussion focuses on VitD, but it was not measured in this study.

L 328. Do no start a new paragraph with Therefore.

L 401. Another limitation of the study is that the physiological implications of low blood calcium were not explored.

Tables

Remove the table’s superscripts indicating statistical significance (no needed). Explain in tables and manuscript if all the variables selected to be in multivariable model based on p-value, remained in the model or were removed after backward elimination.

The tables should stand on their own. Include the definition of hypocalcemia (lab method and threshold) used in the data analysis.

7. PLOS authors have the option to publish the peer review history of their article (what does this mean?). If published, this will include your full peer review and any attached files.

Reviewer #1: No

---

## [Author Response · Author response to Decision Letter 0]

22 Jan 2022

General reviewers comments 

 We wish to appreciate the efforts and work done by the reviewer to strengthen this manuscript. The comments raised by the reviewer have gone a long way to significantly improve the quality of the manuscript. The response to the different reviewer’s comments are presented below.

This manuscript explores associations of low blood Ca in pregnant women and various sociodemographic, obstetric or nutritional factors. This cross-sectional study was conducted in 4 different health care settings in Cameroon. 

Resp: Thanks for the observation.

Consider using the term low blood Ca in clinically healthy subject throughout the manuscript, instead of hypocalcemia. Please note that the title reads low blood Ca. Hypocalcemia connotates an abnormality with potentially negative clinical effects. 

Resp: Thanks for your suggestion. Hypocalcaemia (generally used to mean “lower than normal blood calcium levels”) can either be an asymptomatic laboratory finding or a life-threatening metabolic disturbance (1)(2). We however agree on the precision “in clinically healthy or “apparently” healthy pregnant women” which has been used throughout the manuscript and made clear in the conclusion. This correction has been integrated in the whole manuscript where necessary.

No reference has been cited to support the choice of low blood Ca/hypocalcemia thresholds.

Resp: Thanks for the observations. The reference has been included. See page 10, first paragraph

There is a large disagreement in prevalence of low blood Ca, based on the type of the analytical method of choice. This issue deserves further attention. Please consider the following:

1. Visually evaluate your data by plotting iCa (consider the gold standard) vs total calcium, and iCa vs albumin-corrected total Ca.

2. Evaluate the correlation of iCa with tCa and albumin-corrected tCa.

3. Consider evaluating if there is a bias across lab methods, and if this bias is constant.

Resp: Thank you for your observations and suggestions. These were some original worries when the protocol was been designed. We intended to use ionised hypocalcaemia as a gold standard to evaluate the sensitivity of total calcaemia measurements in defining hypocalcaemia. We however found out that this was not scientifically correct given the ionised and total calcium are not exactly the same laboratory finding diagnosed by different methods. Multiple studies have agreed that even though total hypocalcaemia seems highly prevalent in pregnancy (laboratory finding), this constitutes some form of false or factitious hypocalcaemia associated with physiological changes in pregnancy that can be ignored (3). Ionised and total hypocalcaemia in pregnancy should therefore be studied as two entities as they might be associated with adverse outcomes of different severities.

However, the results of the requested additional analyses have been presented in the results section. See the two attached figures and page 14, paragraph 2 and page 15

Given that we are measuring different outcomes with different methods, we think that it will not make sense to assess bias across methods. Besides some studies suggest that different health conditions affect these different forms of calcium differently.

The criterion used to stratify the sociodemographic (i.e., age, monthly revenue, household occupants), obstetric (i.e., gestational age at first antenatal visit, number of visits, number of pregnancies, number of deliveries), and nutritional factors (i.e., taking fruit rare) should be clear. Describe the reasoning behind the choice of categories 

Resp: Thank you for your question. Given that we did not find any similar studies carried out with categorization. Most of the categorization was done from previous studies carried out in pregnancy and particular WHO health indicators. Some of the variables were categorized following standard divisions like in a union/not in a union.

Concerning fruit consumption, people who consumed fruits rarely were expected to have hypocalcaemia compared with their counterparts who do more often (4).

See paragraph 2, page 10.

Authors do not provide enough background to understand the selected nutritional factors as explanatory variables within the context of Cameroon dietarian preferences. For example, it is unclear why authors decided on oranges, watermelons, and pineapples to classify the most consumed fruit. “Eating fruit rarely” it is not a quantifiable variable and may have different meaning across countries or ethnicities (i.e., number of pieces or servings per day). 

Resp: Thanks for your suggestion. Studies around nutritional habits and practices in Cameroon are still very few. Studies carried out in pregnant women in Cameroon are ever rarer. We did not have any published studies on dietarian practices. 

Concerning fruits, Consumption of vegetables and fruits has been reported to better calcium availability in blood and bone mineral density. We therefore believed that women who consumed fruits rarely were likely to have lower calcaemia compared to their counterparts who consumed fruits more often (4).

We recognize the fact that this variable is not quantifiable and will include this in the limits of the study. In our context, it is really difficult to evaluate this in pieces or servings per day). See limits section, page 30, paragraph 2.

As concerns the most consumed fruits, only pineapples, oranges, watermelons and ripe banana were specified in the questionnaire and all other fruits included into others. When analysed without categories, no particular fruit had a statistically significant association with hypocalcaemia. As stated in the background, the most consumed fruits were watermelon, oranges and pineapples. We therefore wanted to know if consumption of the most frequently consumed fruits in Cameroon was associated with hypocalcaemia. Some information on the nutritional behaviors has been added to the background. See page 5, paragraph 3.

Specific Reviewers comments Response to comments

L 1. Acknowledge that the study was conducted in Cameroon. 

Resp: Thanks for the recommendation. Cameroon has been added to the title. See line 1

L164. As indicated above, authors need to support the hypocalcemia thresholds with references (iCa, albumin-corrected tCa, and tCa). 

Resp: Thanks for your recommendations, the references have been added. Already addressed above

L 164. Report the number of cows identified with low Ca serum when combining result from the 3 methods. 

Resp: Thank you for the recommendation. The number of women meeting the criteria has been added. Only 28 participants met this criteria. See results section, page 14, paragraph 1.

L 202. Please explain in materials how data was analyzed to calculate the prevalence of low calcium in blood. 

Resp: Thank you for the suggestion. The cut-offs considered were stated clearly in the data analysis section. All women who had ionised, total crude and albumin-corrected total calcium levels less than or equal to the stated cut-offs were classified as hypocalcaemic (low blood calcium). This statement has been included in the data analysis section. See page 10, paragraph 1.

L 217. Consider rephrasing as “Sociodemographic and obstetric factors associated with hypocalcemia are presented in Table 3.”

Resp: Thank you for your correction. Adopted in the text. See page 16, paragraph 1.

L 236. Change “Table 8: Nutritional factors affected” by “Table 4: Nutritional factors associated”.

Resp: Thank you for the correction. Corrected accordingly. See table 4, page 19.

L 250. The first discussion paragraph is written as results. Follow scientific writing guidelines for discussion, and delete it. 

Resp: Thank you for your recommendation. It has been deleted and reformulated. See paragraph 1, page 21.

L 250. It is striking the difference in prevalence when considering different lab methods. That should be one of the first points of discussion. 

Resp: Thanks for the suggestion. The discussion has been shifted up to start by discussing the discrepancy. The laboratory methods can really not be discussed here given that these are recommended and gold standard procedures especially for ISE potentiometry. The methods adopted (AAS and ISEP) measure two different targets (ionised calcium and total calcium). Ideally ionised (physiological) hypocalcaemia should be the true target and not the total hypocalcaemia, even though having total hypocalcaemia increased the chances of having ionised hypocalcaemia (stated in the discussion). See page 27, paragraph 2. As stated above additional analysis has been presented.

L 260 to 262. Alkalogenic diets (rick in K) may induce pseudohypoparathyroidism. Please see http://doi.org/10.3618/jds.2013-7467

Resp: Thanks for your suggestion. This has been included in the discussion. See page page 28, last paragraph.

L 265. Replace “borderline significance” by “tendency to”. Define in material and methods tendency, for example p< 0.1 

Resp: Thanks for the suggestion. We prefer to state clearly that the association was not statistically significant. And include the tendency in the next sentence. See page 30, paragraph 1.

L 280. What was the prevalence with unadjusted Calcium levels? 

Resp: Thank you for the question. The prevalence with unadjusted calcium levels is the prevalence of totalcrude hypocalcaemia. That is, when the total calcium levels are not adjusted or corrected for albumin changes. This has been reported in the results and first paragragh of the discussion. The word “unadjusted” has been changed and made more explicit. See page 24, paragraph 2.

L 296 to 299. Disagreement in Ca concentration across lab methods has been reported in patients with blood Ca metabolism disorders. However, the present study included healthy patients. Are there other plausible explanations? As indicated above, please explore the data. 

Resp: Thanks for your concern. As already explained above, only two diagnostic methods were used and these two diagnostic methods did not measure the same calcium fractions in blood. One measured and classified calcium levels based on the physiologically active fraction which is the absolute hypocalcaemia, while the other method measured the whole calcium in circulation (which can be considered a relative hypocalcaemia). See page 21, paragraph 2, sentence 3 and 4, of discussion

L 306. Replace regnancies with pregnancies.

Resp: Thank you for the correction. This has been corrected. See page 22, first paragraph. 

L 320. This section of the discussion focuses on VitD, but it was not measured in this study.L 328. Do no start a new paragraph with Therefore. 

Resp: Thanks for the observation. Vit D deficiency and other hormonal disorders were not measured in this study (stated as a limit). The discussion on these was just to identify based on relative prevalence from other studies, the plausible causes of the hypocalcaemia (particularly the ionised hypocalcaemia) in pregnancy. The “therefore” has been taken off the sentence. See page 23, paragraph 2

L 401. Another limitation of the study is that the physiological implications of low blood calcium were not explored. 

Resp: Thanks for the suggestion. This has been included as a limit. See page 30, last paragragh.

 Tables

Remove the table’s superscripts indicating statistical significance (no needed). Explain in tables and manuscript if all the variables selected to be in multivariable model based on p-value, remained in the model or were removed after backward elimination.

The tables should stand on their own. Include the definition of hypocalcemia (lab method and threshold) used in the data analysis.

Resp: Thanks for the recommendations. The tables have been edited accordingly. See tables 3, 4 and 5.

---

## [Decision Letter · Decision Letter 1]

14 Feb 2022

PONE-D-21-31669R1Low blood calcium in pregnancy: prevalence and risk factors in a resource-limited setting, CameroonPLOS ONE

Dear Dr. Ajong,

Thank you for submitting your manuscript to PLOS ONE. After careful consideration, we feel that it has merit but does not fully meet PLOS ONE’s publication criteria as it currently stands. Therefore, we invite you to submit a revised version of the manuscript that addresses the points raised during the review process.

We look forward to receiving your revised manuscript.

Kind regards,

Frank T. Spradley

Academic Editor

PLOS ONE

Journal Requirements:

Reviewers' comments:

Reviewer's Responses to Questions

**Comments to the Author**

1. Does the manuscript adhere to the experimental procedures and analyses described in the Registered Report Protocol?

If the manuscript reports any deviations from the planned experimental procedures and analyses, those must be reasonable and adequately justified.

Reviewer #1: Yes

2. If the manuscript reports exploratory analyses or experimental procedures not outlined in the original Registered Report Protocol, are these reasonable, justified and methodologically sound?

A Registered Report may include valid exploratory analyses not previously outlined in the Registered Report Protocol, as long as they are described as such.

Reviewer #1: Yes

3. Are the conclusions supported by the data and do they address the research question presented in the Registered Report Protocol?

The manuscript must describe a technically sound piece of scientific research with data that supports the conclusions. The conclusions must be drawn appropriately based on the research question(s) outlined in the Registered Report Protocol and on the data presented.

Reviewer #1: Yes

4. Have the authors made all data underlying the findings in their manuscript fully available?

Reviewer #1: Yes

5. Is the manuscript presented in an intelligible fashion and written in standard English?

Reviewer #1: No

6. Review Comments to the Author

Please use the space provided to explain your answers to the questions above. (Please upload your review as an attachment if it exceeds 20,000 characters)

Reviewer #1: Thanks to the authors for revising the manuscript. It has greatly improved. I have additional comments that may help improve the quality of this work.

Consider that Plos One does not copyedit manuscripts. The language used is correct but not very elegant. I will strongly recommend a professional editor to polish the manuscript prior to publication.

Major: Why the study focuses on hypocalcemia based on total Ca (tCa) instead of ionized Ca (iCa)? Should be more appropriate to present associations with both iCa and tCa? The number of subjects with low iCa is limited but enough to explore possible associations.

Minor: Revise the manuscript and use concentration instead level. Be consistent.

Specific comments:

L 34. Please consider rephrasing: Blood samples were collected and analyzed for serum ionised calcium concentrations and pH (ion-selective potentiometry), and for total calcium and albumin concentration (atomic absorption spectrophotometry). Sociodemographic, obstetric and nutritional data were collected through interviewer-administered questionnaire.

L57 – 66. Indicate the type of analysis (iCa or tCa) used to evaluate hypocalcemia in studies 1 to 5.

L 167. Explain in this section the software used to do the statistical analysis, and methods for regression and logistic regression.

L 167. Explain in the data analysis section what measurement and definition of hypocalcemia was used to build the multivariable models.

L 248 to 251. Is this information a repetition of what presented in Fig 1 and 2? If so, delete.

L 259. Consider using the word odds when reporting results. For example “Participants who earned less than 100.000FCFA (179 USD) in a month had 0.73 lower odds to have total crude hypocalcaemia”. Revise entire manuscript. This is important because this study reports odds no likelihood or risks.

L 308 discusses de physiological implications of iCa but the study evaluates associations with tCa.

L 293 to 295. This is results. It does not belong in the discussion.

Table 3 and 4 footnotes, modified as:

p-value <0.05; statistical significance

p-value<0.25; eligible for multiple regression analysis

7. PLOS authors have the option to publish the peer review history of their article (what does this mean?). If published, this will include your full peer review and any attached files.

Reviewer #1: No

---

## [Author Response · Author response to Decision Letter 1]

19 Feb 2022

RESPONSE TO REVIEWERS

We wish to thank the editor and reviewer for the time taken to improve on this manuscript. We have now taken time to address all the editor and reviewer comments. This is presented below. 

Editor’s comments

Comment: Please review your reference list to ensure that it is complete and correct. If you have cited papers that have been retracted, please include the rationale for doing so in the manuscript text, or remove these references and replace them with relevant current references. Any changes to the reference list should be mentioned in the rebuttal letter that accompanies your revised manuscript. If you need to cite a retracted article, indicate the article’s retracted status in the References list and also include a citation and full reference for the retraction notice.

Response: Thanks for your advice. The references have been checked and some changes made. Ref number 8 and 11 have been changed, and one of the references which was a repetition of reference number 3 taken off. See reference list.

Reviewer’s comments

Comment: Thanks to the authors for revising the manuscript. It has greatly improved. I have additional comments that may help improve the quality of this work.

Response: Thanks for your observation. The efforts put in by the reviewer to improve this work are highly appreciated on our side. We must agree that the review process has highly improved on the quality of this manuscript. 

Comment: Consider that Plos One does not copyedit manuscripts. The language used is correct but not very elegant. I will strongly recommend a professional editor to polish the manuscript prior to publication.

Response: Thanks for the observation. We had our manuscript screened by a native English speaker and hope the level of English is up to standard now.

Major comment: Why the study focuses on hypocalcemia based on total Ca (tCa) instead of ionized Ca (iCa)? Should be more appropriate to present associations with both iCa and tCa? The number of subjects with low iCa is limited but enough to explore possible associations.

Response: We highly appreciate the reviewers question which is pertinent. We belief that leaving this out would have taken a big chunk off this work. As recommended, we have added analysis for ionised calcaemia and changes made on the whole manuscript to reflect the analyses. This has affected the objective statement, the results, discussion and conclusion. See abstract (page 2 and 3), last paragraph of introduction (page 6), results section (page 21, 22 and 24, including table 5, 6 and 7), discussion section (page 32, second paragraph and page 33, last paragraph), conclusion section, first paragraph, page 35).

Minor comment: Revise the manuscript and use concentration instead level. Be consistent. 

Response: Thanks for your correction. The whole manuscript has been revised as recommended.

Specific comments:

Comment: L 34. Please consider rephrasing: Blood samples were collected and analyzed for serum ionised calcium concentrations and pH (ion-selective potentiometry), and for total calcium and albumin concentration (atomic absorption spectrophotometry). Sociodemographic, obstetric and nutritional data were collected through interviewer-administered questionnaire.

Response: Thanks for taking time to correct this. We have adopted it as recommended. See Abstract, page 2, line 35-39.

Comment: L57 – 66. Indicate the type of analysis (iCa or tCa) used to evaluate hypocalcemia in studies 1 to 5.

Response: Thanks for your suggestion. This has been added to the first paragraph of the introduction.

Comment: L 167. Explain in this section the software used to do the statistical analysis, and methods for regression and logistic regression.

Response: Thanks for your suggestion: All these are included in the data analysis section. See page 9, 10 and 11.

Comment: L 167. Explain in the data analysis section what measurement and definition of hypocalcemia was used to build the multivariable models.

Response: Thanks for your suggestion. This has been included in the data analysis section. See page 10, paragraph 2.

Comment: L 248 to 251. Is this information a repetition of what presented in Fig 1 and 2? If so, delete.

Response: Thanks for your observation. This is not a repetition but additional information not accessible on the figures. We think the information should remain.

Comment: L 259. Consider using the word odds when reporting results. For example “Participants who earned less than 100.000FCFA (179 USD) in a month had 0.73 lower odds to have total crude hypocalcaemia”. Revise entire manuscript. This is important because this study reports odds no likelihood or risks.

Response: Thank you very much for pulling our attention to that. We have revised the whole manuscript accordingly. 

Comment: L 308 discusses de physiological implications of iCa but the study evaluates associations with tCa.

Response: Thanks for your observation. We have now included associations with iCa in the manuscript.

Comment: L 293 to 295. This is results. It does not belong in the discussion.

Thanks very much, it has been taken up to the results section. See result section, page 16, line 286-288.

Comment: Table 3 and 4 footnotes, modified as:

p-value <0.05; statistical significance

p-value<0.25; eligible for multiple regression analysis

Response: Thanks for your suggestion. This has been modified.

---

## [Decision Letter · Decision Letter 2]

26 Apr 2022

PONE-D-21-31669R2Ionised and total hypocalcaemia in pregnancy: an analysis of prevalence and risk factors in a resource-limited setting, CameroonPLOS ONE

Dear Dr. Ajong,

Thank you for submitting your manuscript to PLOS ONE. After careful consideration, we feel that it has merit but does not fully meet PLOS ONE’s publication criteria as it currently stands. Therefore, we invite you to submit a revised version of the manuscript that addresses the points raised during the review process.

We look forward to receiving your revised manuscript.

Kind regards,

Frank T. Spradley

Academic Editor

PLOS ONE

Journal Requirements:

Reviewers' comments:

Reviewer's Responses to Questions

**Comments to the Author**

1. Does the manuscript adhere to the experimental procedures and analyses described in the Registered Report Protocol?

If the manuscript reports any deviations from the planned experimental procedures and analyses, those must be reasonable and adequately justified.

Reviewer #1: Yes

2. If the manuscript reports exploratory analyses or experimental procedures not outlined in the original Registered Report Protocol, are these reasonable, justified and methodologically sound?

A Registered Report may include valid exploratory analyses not previously outlined in the Registered Report Protocol, as long as they are described as such.

Reviewer #1: Yes

3. Are the conclusions supported by the data and do they address the research question presented in the Registered Report Protocol?

The manuscript must describe a technically sound piece of scientific research with data that supports the conclusions. The conclusions must be drawn appropriately based on the research question(s) outlined in the Registered Report Protocol and on the data presented.

Reviewer #1: Yes

4. Have the authors made all data underlying the findings in their manuscript fully available?

Reviewer #1: Yes

5. Is the manuscript presented in an intelligible fashion and written in standard English?

Reviewer #1: Yes

6. Review Comments to the Author

Please use the space provided to explain your answers to the questions above. (Please upload your review as an attachment if it exceeds 20,000 characters)

Reviewer #1: 

Thank you for your effort. Please see below some minor comments.

L 254 to 257 Please note that your sample size is very large and results in significant differences, but the correlation coefficient is poor.

L 262 to 263 Your sample size is very large and results in significant differences, but it may have limited biological implications.

L 332. Replace “Piece of work” with “The main goal of our study was to estimate the prevalence of …”

All the figures and tables should stand on their own without the need to gather additional information from the main body of the manuscript. Add the number of subjects in the total and ionized hypocalcemia. Please revise the space around equal signs in the tables and figures.

7. PLOS authors have the option to publish the peer review history of their article (what does this mean?). If published, this will include your full peer review and any attached files.

Reviewer #1: No

---

## [Author Response · Author response to Decision Letter 2]

27 Apr 2022

RESPONSE TO REVIEWERS

We appreciate the immense effort of the editor and the reviewer towards strengthening this work. We have addressed the comments raised by the editor and reviewer below.

Editor’s comment: Please review your reference list to ensure that it is complete and correct. If you have cited papers that have been retracted, please include the rationale for doing so in the manuscript text, or remove these references and replace them with relevant current references. Any changes to the reference list should be mentioned in the rebuttal letter that accompanies your revised manuscript. If you need to cite a retracted article, indicate the article’s retracted status in the References list and also include a citation and full reference for the retraction notice.

Response: Thanks for the recommendation. The reference list had been reviewed accordingly during the last revision and the changes were specified in the rebuttal letter.

Reviewer’s comments

Comment: Thank you for your effort. Please see below some minor comments.

Response: Thank you again for additional comments. We are grateful for you pertinent contributions

Comment: L 254 to 257 Please note that your sample size is very large and results in significant differences, but the correlation coefficient is poor.

Response: Thank you for the observation. We have taken this precision off given that it is not part of the objectives of this manuscript. However, we feel that not all statistical observations have direct clinical or biological implications. We already stated in the manuscript that the correlation was weak (poor). See line 254-257.

Comment: L 262 to 263 Your sample size is very large and results in significant differences, but it may have limited biological implications.

Response: Thank you for the comment; given that this is not part of any of the objectives of the manuscript, the specification has been removed. See results section, line 262-267.

Comment: L 332. Replace “Piece of work” with “The main goal of our study was to estimate the prevalence of …”

Response: Thank you for your suggestion. It has been adopted. See discussion section, line 332. 

Comment: All the figures and tables should stand on their own without the need to gather additional information from the main body of the manuscript. Add the number of subjects in the total and ionized hypocalcemia. Please revise the space around equal signs in the tables and figures.

Response: Thank you for your comments; we have now made the requested additions to the tables and manuscript.

---

## [Editor Report · Decision Letter 3]

5 May 2022

Ionised and total hypocalcaemia in pregnancy: an analysis of prevalence and risk factors in a resource-limited setting, Cameroon

PONE-D-21-31669R3

Dear Dr. Ajong,

We’re pleased to inform you that your manuscript has been judged scientifically suitable for publication and will be formally accepted for publication once it meets all outstanding technical requirements.

Kind regards,

Frank T. Spradley

Academic Editor

PLOS ONE

---

## [Editor Report · Acceptance letter]

10 May 2022

PONE-D-21-31669R3 

Ionised and total hypocalcaemia in pregnancy: an analysis of prevalence and risk factors in a resource-limited setting, Cameroon 

Dear Dr. Ajong:

I'm pleased to inform you that your manuscript has been deemed suitable for publication in PLOS ONE. Congratulations! Your manuscript is now with our production department. 

Kind regards, 

on behalf of

Dr. Frank T. Spradley 

Academic Editor

PLOS ONE